# A Method for Estimating Time-Dependent Corrosion Depth of Carbon and Weathering Steel Using an Atmospheric Corrosion Monitor Sensor

**DOI:** 10.3390/s19061416

**Published:** 2019-03-22

**Authors:** Jin-Hee Ahn, Young-Soo Jeong, In-Tae Kim, Seok-Hyeon Jeon, Chan-Hee Park

**Affiliations:** 1Department of Civil Engineering, Gyeongnam National University of Science and Technology, 33 Dongjin-ro, Jinju, Gyeongnam 52725, Korea; jhahn@gntech.ac.kr (J.-H.A.); jshyeon1950@gmail.com (S.-H.J.); 2Seismic Research and Test Center, Pusan National University, Yansan, Gyeongnam 50612, Korea; 3Department of Civil Engineering, Pusan National University, San 30, Jangjeon-dong, Geumheong-gu, Busan 46241, Korea; itkim@pusan.ac.kr; 4Steel Structure Research Group, 100 Songdogwahak-ro, Yeonsu-gu, Incheon 21985, Korea; chanhee.park@posco.com

**Keywords:** uncoated carbon steel, weathering steel, corrosion depth, ACM-type sensor, acceleration corrosion test

## Abstract

In this study, a time-dependent corrosion depth estimation method using atmospheric corrosion monitor (ACM) sensor data to evaluate time-dependent corrosion behaviors is proposed. For the time-dependent corrosion depth estimation of uncoated carbon steel and weathering steel, acceleration corrosion tests were conducted in salt-spray corrosion environments and evaluated with a corrosion damage estimation method using ACM sensing data and corrosion loss data of the tested steel specimens. To estimate the time-dependent corrosion depth using corrosion current by an ACM sensor, the relationship between the mean corrosion depth calculated from the weight loss method and the corrosion current was evaluated. The mean corrosion depth was estimated by calculating the corrosion current and evaluating the relationship between the mean corrosion depth and corrosion current during the expected period. From the test and estimation results, the corrosion current demonstrated a good linear correlation with the mean corrosion depth of carbon steel and weathering. The calculated mean corrosion depth is nearly the same as that of the tested specimen, which can be well used to estimate corrosion rate for the uncoated carbon steel and weathering steel.

## 1. Introduction

Steel structures that function as structural members of infrastructure or social structures are generally exposed to corrosive atmospheric environments, including humidity, temperature, airborne salt from the sea, and chemical components in certain areas. These atmospheric corrosive environments result in corrosion problems in steel structures owing to the oxidation characteristics of structural steel materials. Corrosion damage on the surface of steel structures can produce severe structural problems, such as the collapse of plate girder bridges, truncation of truss bridge members, or fatigue cracking of orthotropic bridge decks which have all been reported to degrade structural performance or cause failure [1,2]. Therefore, it is useful to estimate the time-dependent corrosion damage on the surface of a steel structure to prevent severe corrosion damage and develop corrosion maintenance methods based on the independent corrosion environments of steel structures. However, it is difficult to clarify the time-dependent corrosion damage quantitatively based on independent corrosion environments and steel material conditions, such as steel grade or type, as determined by chemical components. To address these problems, the corrosion losses of steel materials can be monitored by a sensor system. Impedance-based [3,4,5,6] and atmospheric corrosion monitor (ACM) [7,8,9,10] sensors using electrochemical signals have been developed, and electrochemical techniques have been used widely to monitor the corrosion rate of structures and the corrosivity of corrosion environments [11]. The corrosion rate and corrosion blister of coating layers have been evaluated using pulsed eddy currents [12,13]. Fe/Ag galvanic-coupled ACM sensors can be used to evaluate the corrosion environments and corrosion conditions of steel plates [14,15,16]. However, it is not clearly established that the corrosion environments can be evaluated considering the rainfall effect and the corrosion delay effect causing rust layers for each steel material using the ACM sensors. Thus, an evaluation method of the time-dependent corrosion behaviors of carbon steel plates was suggested based on the amount of daily average electricity and considering the ACM sensor output [14,15,16]. Further, a method of estimating the time-dependent corrosion depth of a carbon steel plate was proposed that considers various corrosion environments determined by regional characteristics, such as rainfall and airborne salt [13]. However, the estimation method for time-dependent corrosion depth can differ according to the steel material. Particularly, weathering steel, which is generally used for structural members of infrastructures without anticorrosion coatings, can be evaluated for corrosion damage differently using the proposed corrosion depth estimation method. Previously, the estimation of the time-dependent corrosion depths of carbon steel has been examined experimentally from the corrosion product layer [17,18].

In this study, a time-dependent corrosion depth estimation method for carbon steel and weathering steel is proposed using the corrosion current obtained from an ACM sensor affected by the corrosion environment based on comparing the time-dependent corrosion depths. Accelerated corrosion tests for carbon steel and weathering steel were conducted in low salt-spray corrosive environments and the corrosion currents were measured using an ACM sensor during the test periods. From the measured corrosion currents and acceleration corrosion tests, the corrosion current was calculated from the ACM sensor and compared with the mean corrosion depth of the tested specimen. The corrosion current demonstrated a good linear correlation with the mean corrosion depth of carbon steel and weathering steel. The real mean corrosion depths of the specimen were compared to the calculated mean corrosion depths.

## 2. Prediction Method for Mean Corrosion Depth Based on Corrosion Current

The method of predicting the corrosion rate consists of an electrochemical method and a weight loss method. An electrochemical measurement made by an ACM sensor relates the galvanic current to the corrosion rate. ACM sensors have been applied to monitor corrosion in industrial plants and infrastructures. A linear relationship between the atmospheric corrosion rate and sensor galvanic current output has been suggested for severe marine and rural environments [19]. However, to obtain the corrosion rate under an atmospheric environment, the test periods required are long and the economic costs are high. In this study, the mean corrosion depths were estimated using the corrosion current and weight loss in less time.

### 2.1. Measurement of Corrosion Current Using an ACM Sensor

An ACM sensor is composed of two different metal electrodes that are initially insulated from each other, as shown in Figure 1. When a thin water film is generated on the insulator and the two electrodes by dew condensation, the two electrodes conduct electrically, and a galvanic current flows in the sensor. The galvanic current is related to the corrosion rate and the conductance of the water film by dew condensation. It indicates a high correlation with the corrosion rate of actual metals [9]. However, the galvanic current is known to be larger than the microcell corrosion current of steel during rainfall [15]. Moreover, the evaluation for the corrosion rate of uncoated carbon steel and weathering steel should be considered based on the effect of rainfall and formed corrosion products on steel surface. Because of this problem, the current cannot be easily used to compare the corrosion rate of steel. In this study, the error of the corrosion current caused by heat conduction between the ACM sensor and tested specimen was minimized using a thermally conductive sheet. In addition, the start and end of rainfall from the corrosion current by the ACM sensor were defined to consider the rainfall effect. The rain start was observed over 1 μA, and the rain ended when the rate of change for the corrosion current from the start of rainfall became 1/2 times or less than 1 μA after 10 min. The corrosion currents from the ACM sensor (output: 0.1 nA to 1 mA, resolution: 0.1 nA (0.1 nA to 10 μA), 1 μA (1 μA to 10 mA)) were measured every 10 min for the carbon steel plate specimens.

### 2.2. Prediction Method of Corrosion Damage

To evaluate the anticorrosion performance of steel, the most reasonable method is to study the deterioration rate of steel in an atmospheric corrosive environment depending on the exposure period. However, the anticorrosion performance of an actual steel structure is limited by the irregularity of the corrosion rate and deterioration pattern in the installed environment and conditions. The prediction of corrosion rate using corrosion current tends to overestimate the corrosion depth of the corrosion damage. The corrosion currents increase linearly with increasing test time and the corrosion damage decreases with increasing test time in a constant corrosion environment. Therefore, in the method of predicting the mean corrosion depth using an ACM sensor, the mean corrosion depth was calculated using the relationships of corrosion depth and corrosion current under the same corrosion environment.

The prediction method was processed from the first to the fourth step, as shown in Figure 2. First, the corrosion damage was evaluated using the weight loss method depending on the exposure period by acceleration corrosion test. Next, the output of the ACM sensor was converted to consider environmental effects such as water vapor, and the accumulated corrosion current was calculated using the ACM sensor. Subsequently, the relationship equation was used to compare the mean corrosion depth and corrosion current in the same corrosion environment and exposure period. Finally, the mean corrosion depth was estimated by calculating the corrosion current during the expected period.

## 3. Accelerated Corrosion Test and Results

### 3.1. Acceleration Corrosion Test Condition and Results

To review and compare the proposed corrosion damage prediction method, as shown in Figure 3, we conducted accelerated corrosion test [20]. In the conducted accelerated corrosion tests [21], uncoated carbon steel and weathering steel plates were applied to evaluate the corrosion damage according to the JWTCS1001 specification. The JWTCS1001 test method evaluates the corrosion resistance of low-carbon steel in an atmospheric environment with relatively low airborne salt. 

The test method was combined in the Phase-A cycle with salt spray and the Phase-B cycle without salt spray according to the specification [21]. The salt accumulation on the surface of the specimen in the Phase-A cycle, and, in the Phase-B cycle, corrosion was accelerated to repeat the dry and wet conditions. The Phase-A cycle consisted of 0.5-h wet periods under air stream at 35 °C and relative humidity 90%, 0.5-h fog cycle using a solution of 1 wt.%, 1-h wet periods under air stream at 35 °C and relative humidity 90%, and 6.0-h dry periods under air stream at 40 °C and relative humidity 50%. The Phase-B cycle consisted of 2.0-h wet periods under air stream at 35 °C and relative humidity 90%, and 6.0-h dry periods under air stream at 35 °C and relative humidity 90%. Accelerated corrosion tests were conducted repeatedly in the Phase-A cycle for 144 h and subsequently in the Phase-B cycle for 96 h in a Q-FOG CRH chamber. The corrosive environment conditions applied during the accelerated corrosion tests are summarized in Figure 3a. The accelerated corrosion tests were performed for 10, 30, 50, and 100 days [20]. Steel plate specimens of 400 mm length, 60 mm width, and 9 mm thickness were constructed using the Korea Industrial Standard (KS) SM355A and HSB380W structural steel [22]. The focus surface of the specimens was subjected to a blast surface treatment (ISO 8503 Sa2.5), and the other side was coated with a silicon component for anticorrosion purposes. The mean corrosion depths were calculated using the weight loss method after rust removal from each specimen. Figure 3b shows the setup of the accelerated corrosion tests [20].

In general, the mean corrosion depth was calculated based on weight loss, and the evaluation of the relationship between the mean corrosion depth and exposure period was used in a power function, as in Equation (1) [23]:(1)$$Y=A\cdot X^{B},$$where Y is the mean corrosion depth (mm), X is the exposure period (years), and A and B are constants.

The mean corrosion depth after 10, 30, 50, and 100 days using the weight loss method is shown in Figure 4 [20]. The relation coefficient values were obtained from Equation (1). The mean corrosion depth increased with exposure period, but the amount of increase during each exposure period decreased. This could be because extremely compact rust layers were formed, thereby providing protection against corrosion damage.

### 3.2. Corrosion Current Using an ACM Sensor

When a thin water film forms on a steel surface, the ACM sensor interprets it as rainfall or dew, and a galvanic current passes between the substrate (Fe) and conductive paste (Ag). It has been reported that this galvanic current demonstrates a good correlation with the corrosion rate of metals, such as steel and zinc. Thus, by measuring the ACM sensor output, the corrosion environments can be evaluated quantitatively. The ACM sensor output for rainfall was greater than 10 μA (rainfall threshold), and that for dew was greater than 0.1 μA (dew threshold) [11]. The corrosion current was measured by the ACM sensor during the acceleration corrosion test, as shown in Figure 5. In the Phase-A cycle, the corrosion current increased in wet conditions and increased significantly after a salt spray. The corrosion current was greater than 10 μA for the rainfall threshold. However, the corrosion current of the Phase-B cycle increased in wet conditions and decreased in dry conditions. The currents in the Phase-A cycle were larger than those in the Phase-B cycle, which is likely caused by the difference in the amount of salt that adhered to the surfaces.

## 4. Estimated Time-Dependent Mean Corrosion Depth

For a quantitative analysis, the corrosion current was calculated from the ACM sensor outputs and compared with the corrosion damage of the tested specimens. Figure 6 shows the relationship between the mean corrosion depth and corrosion current of SM355A and HSB380W. The corrosion rate tends to decrease as the corrosion current increases. The corrosion current demonstrated a good linear correlation with the mean corrosion depth of carbon steel and weathering steel. Equations (2) and (3) were obtained from the linear regressions for 10, 30, and 50 days.(2)$$d_{mean}=102.0\cdot \log\left(A\right)-162.1\;\mathrm{for}\;\mathrm{SM}355A,$$
(3)$$d_{mean}=76.6\cdot \log\left(A\right)-105.9\;\mathrm{for}\;\mathrm{HSB}380W,$$

To examine the prediction method for the mean corrosion depth of uncoated carbon steel and weathering steel, the real and calculated mean corrosion depths of the 100-day specimen were compared. Figure 7 shows the tested and predicted mean corrosion depth of each steel grade. The calculated mean corrosion depth is nearly the same as that of the tested specimen, as shown in Figure 8.

In addition, the long-term corrosion depths were calculated under the measured corrosion current electricity in the accelerated corrosion tests. Figure 9 shows the mean corrosion depth that was predicted by the ACM for 1 to 10 years from the calculated long-term corrosion depth of each steel specimen under the accelerated corrosion test environment. It is difficult to compare the regression curve of the tests and the prediction curve under a long exposure period, because the amount of corrosion damage decreased with increasing exposure period in a low airborne salt environment and because of the uncertainty of the corrosion progression. However, this estimation method can be useful and effective for the maintenance of steel structures with a corrosion problem according to the environmental corrosion conditions of the installed steel structure. The method can be improved by adding the corrosion damage data and results over time.

## 5. Conclusions

In this study, to suggest an estimation method for the time-dependent corrosion depth of carbon steel and weathering steel, accelerated corrosion tests and measurements of corrosion current using an ACM sensor were conducted. The measured ACM sensor data were evaluated for examining the time-dependent corrosion depth of carbon steel and weathering steel exposed to salt-spray corrosive environments. The corrosion current was evaluated from the ACM sensor and compared with the corrosion damage of the tested specimens. The corrosion rate tended to decrease as the corrosion current increased. The corrosion current demonstrated a good linear correlation with the mean corrosion depth of carbon steel and weathering steel. To examine the prediction method for the mean corrosion depth of uncoated carbon steel and weathering steel, the tested and predicted mean corrosion depths were compared. The calculated mean corrosion depth was nearly the same as that of the tested specimen. In addition, their long-term corrosion depths were calculated for the measured corrosion current in the accelerated corrosion test. In this study, the long-term corrosion depth of steel was estimated from limited test results and conditions. However, this method can provide a reference to maintain steel structures effectively against corrosion damage generated from the atmospheric corrosion environment in an uncertain steel corrosion problem. The research methods discussed in this paper proves that the mean corrosion depths can be forecasted through ACM sensors.

## Figures and Tables

**Figure 1 sensors-19-01416-f001:**
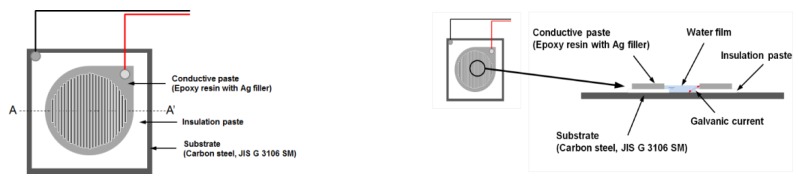
Layout of an ACM sensor.

**Figure 2 sensors-19-01416-f002:**
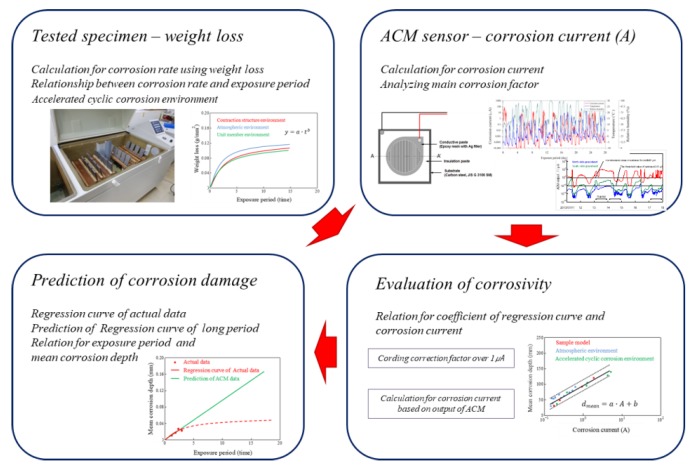
Flow for prediction method by ACM sensor.

**Figure 3 sensors-19-01416-f003:**
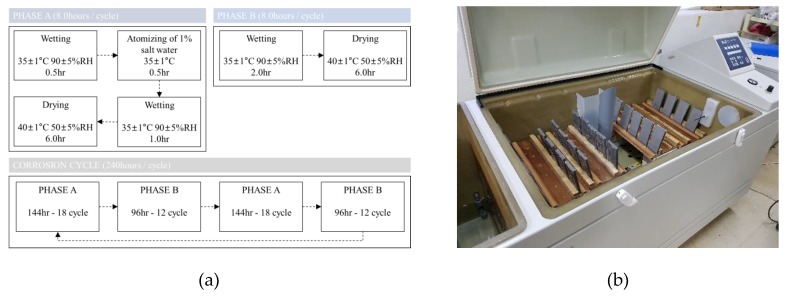
Corrosion cycle tests: (**a**) Test sequence and corrosion conditions of JWTCS1001 [21]; (**b**) Setup of the corrosion cycle test in the Q-FOG CRH chamber [20].

**Figure 4 sensors-19-01416-f004:**
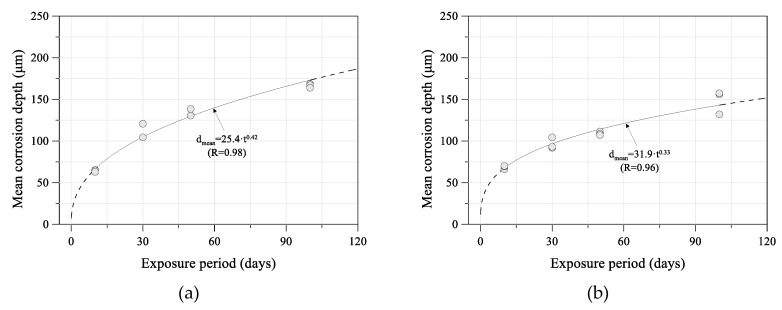
Relationship between mean corrosion depth and exposure period tests: (**a**) uncoated carbon steel (SM355A); (**b**) weathering steel (HSB380W) [20].

**Figure 5 sensors-19-01416-f005:**
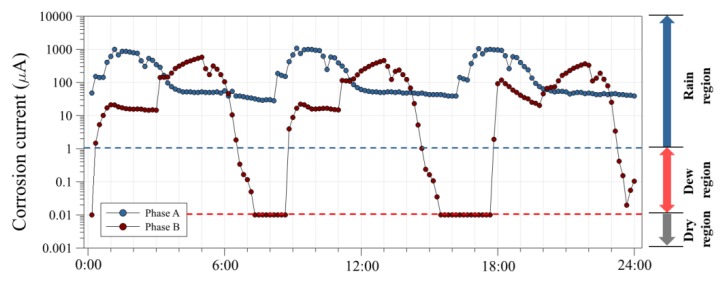
Corrosion current during acceleration corrosion test.

**Figure 6 sensors-19-01416-f006:**
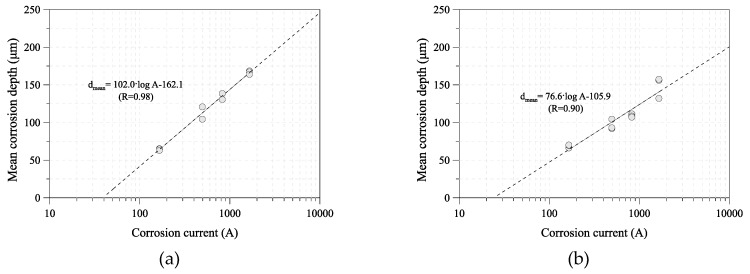
Relationship between mean corrosion depth and corrosion current: (**a**) uncoated carbon steel (SM355A); (**b**) weathering steel (HSB380W).

**Figure 7 sensors-19-01416-f007:**
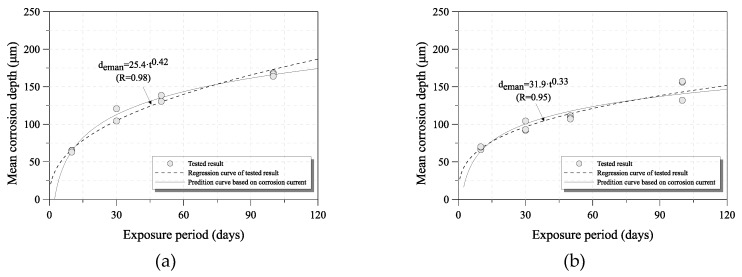
Comparison of real and calculated mean corrosion depths: (**a**) uncoated carbon steel (SM355A); (**b**) weathering steel (HSB380W).

**Figure 8 sensors-19-01416-f008:**
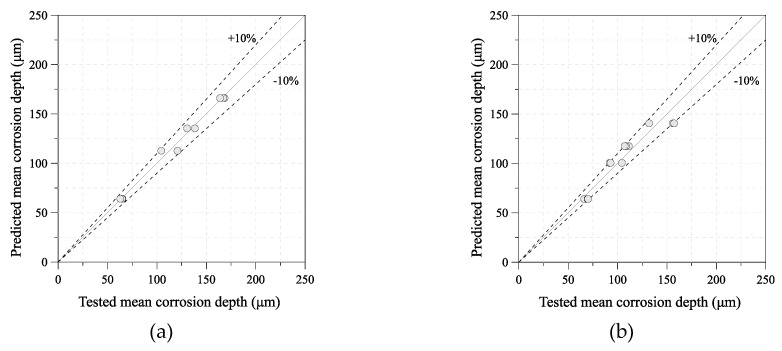
Validation of mean corrosion depth calculated using Equations (2) and (3): (**a**) uncoated carbon steel (SM355A); (**b**) weathering steel (HSB380W).

**Figure 9 sensors-19-01416-f009:**
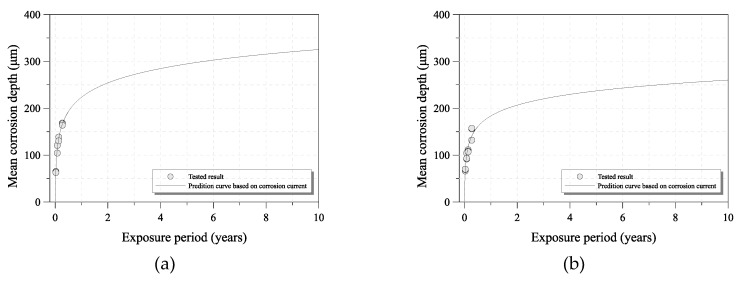
Prediction result of mean corrosion depth under long-term exposure period: (**a**) uncoated carbon steel (SM355A); (**b**) weathering steel (HSB380W).

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
