# Peer review of "A Method for Estimating Time-Dependent Corrosion Depth of Carbon and Weathering Steel Using an Atmospheric Corrosion Monitor Sensor"

_sensors, 2019, doi:10.3390/s19061416_

Round 1
Reviewer 1 Report
exponential function in equation (1) shoub be power function?
the eddy current nondestructive testing could be used for corrosion detection and prediction: Steel corrosion characterization using pulsed eddy current systems
Y He, G Tian, H Zhang, M Alamin, A Simm, P Jackson
IEEE Sensors Journal 12 (6), 2113-2120
and thermography could be used:
An investigation into eddy current pulsed thermography for detection of corrosion blister
Y He, GY Tian, M Pan, D Chen, H Zhang
Corrosion Science 78, 1-6
Author Response
Question 1. Exponential function in equation (1) should be power function?
- As your comment, the manuscript was changed “exponential function” to “power function”.
Question 2. The eddy current nondestructive testing could be used for corrosion detection and prediction: Steel corrosion characterization using pulsed eddy current systems and thermography could be used:
1. “Y He, G Tian, H Zhang, M Alamin, A Simm, P Jackson, IEEE Sensors Journal 12 (6), 2113-2120”
2. “Y He, GY Tian, M Pan, D Chen, H Zhang, Corrosion Science 78, 1-6”
Thank you for the valuable comment, we added the references and manuscript was modified as follows.
“The corrosion rate and corrosion blister of coating layer were evaluated by using pulsed eddy current [12, 13].”

Reviewer 2 Report
The overall presentation of the paper is at a satisfactory level but unfortunately this paper needs a major rewrite for publication in this journal.
My comments are as follows;
1. In the abstract from line 17 to line 24, there is a generic description about time dependent corrosion and its implications on structures. An abstract is a comprehensive reflection about the research undertaken in an article.
2. From line 24 to line 31, there is a descriptive introduction to the research but does not bring forward any results and novelty of the research. Therefor it is important that the abstract is revised.
3. In the introduction section there is a large proportion of literature about the implications of atmospheric corrosion but its lacks relation to the research presented in the paper. Form line 67 to 74 there is a short discussion about the work conducted but unfortunately it is vague.
4. From line 75 [Prediction method for mean corrosion depth based on corrosion current] to line 96, the discussion is changes between atmospheric corrosion to galvanic corrosion and vice versa. There a lack of focus what the authors intend to explain.
5. At line 88 it is mentioned that galvanic corrosion is related to corrosion rate and it states that the galvanic corrosion is larger than microcell corrosion, and then it is mentioned that correlation between galvanic corrosion and corrosion current is different? In its current form section 2.1 is unclear and full of small unconnected sentences.
6. Figure 2 is cluttered with results but what is the setup of the experimental rig? Shown in figure 1.
7. At line 121 [3. Acceleration corrosion test and results], at line 129 it is mentioned that salt spray test were conducted but which one for example ASTM B117? Also salt test are used for aggressive corrosion tests, where is atmospheric corrosion is no an aggressive form of corrosion, unless the air around us has salt a very high salt content. Also the authors didn’t mention how the salt spray and the study of atmospheric corrosion included in this paper are related?
In its current form the paper present some basic literature about corrosion, research methodology and results but unfortunately does not show novelty of the research and could not explain how three different modes of corrosion are related.
Authors are encouraged to improve the quality and resubmit.
Author Response
Question 1. In the abstract from line 17 to line 24, there is a generic description about time dependent corrosion and its implications on structures. An abstract is a comprehensive reflection about the research undertaken in an article.
- As your comment, an abstract must be comprehensive reflection about the research undertaken in an article. The sentence for generic description about time dependent corrosion of steel structure is deleted and the description of test results is added.
Question 2. From line 24 to line 31, there is a descriptive introduction to the research but does not bring forward any results and novelty of the research. Therefor it is important that the abstract is revised.
- As your comment, the manuscript was modified as following.
“To estimate time-dependent corrosion depth using corrosion current by an ACM sensor, relationship between the mean corrosion depth calculated from weight loss method and corrosion current was evaluated. The mean corrosion depth was estimated by calculating corrosion current and evaluating relationship between the mean corrosion depth and corrosion current during the expected period. From the test and estimation results, the corrosion current showed a good linear correlation with the mean corrosion depth of carbon steel and weathering. The calculated mean corrosion depth is nearly the same as that of the tested specimen.”
Question 3. In the introduction section there is a large proportion of literature about the implications of atmospheric corrosion but its lacks relation to the research presented in the paper. Form line 67 to 74 there is a short discussion about the work conducted but unfortunately it is vague.
- As your comment, the manuscript was modified as following.
“In this study, a time-dependent corrosion depth estimation method for carbon steel and weathering steel is proposed using the corrosion current achieved from an ACM sensor affected by the corrosion environment based on comparing the time-dependent corrosion depths. For this purpose, an acceleration corrosion tests for carbon steel and weathering steel were conducted under low salt spray corrosion environments and the corrosion currents were measured by ACM sensor during tested periods. From the measured corrosion currents and acceleration corrosion tests, the corrosion current was calculated from the ACM sensor and compared with the mean corrosion depth of the tested specimen. The corrosion current showed a good linear correlation with the mean corrosion depth of carbon steel and weathering. The real mean corrosion depths of the specimen were compared to the calculated mean corrosion depths.”
Question 4. From line 75 [Prediction method for mean corrosion depth based on corrosion current] to line 96, the discussion is changes between atmospheric corrosion to galvanic corrosion and vice versa. There a lack of focus what the authors intend to explain.
- In general, the mean corrosion depths were predicted based electrical method(sensor) and experimental test method(weight loss). To evaluate the corrosion damage of uncoated carbon steel and weathering steel structure under atmospheric corrosion environments, the measured or exposure period must be need to long-term. In this study, to focus the mean corrosion depths were estimated by acceleration corrosion test in short-term time.
Question 5. At line 88 it is mentioned that galvanic corrosion is related to corrosion rate and it states that the galvanic corrosion is larger than microcell corrosion, and then it is mentioned that correlation between galvanic corrosion and corrosion current is different? In its current form section 2.1 is unclear and full of small unconnected sentences.
- When a thin water film is formed on the surface of the ACM sensor, a galvanic current passes between the substrate (Fe) and the conductive paste (Ag). It has been reported that this galvanic current shows a good correlation with the corrosion rate of actual metals such as steel and zinc. However, during rainfall events, the galvanic corrosion current is larger than microcell corrosion. The manuscript was modified section 2.1 as follow.
“~~. The galvanic current is related to the corrosion rate and the conductance of the water film by dew condensation. Its galvanic current shows that highly correlated with the corrosion rate of actual metals [9]. However, the galvanic current is known to be larger than the microcell corrosion current of steel during rainfall [15]. Moreover, the evaluation for corrosion rate of uncoated carbon steel and weathering steel should be considered in effect of rainfall and formed corrosion products on steel surface. Because of this problem, the current was difficult to compare the corrosion rate of steel. In this study, the error of the corrosion current caused by heat conduction between the ACM sensor and the tested specimen was minimized by using a thermally conductive sheet. In addition, the start and end of rainfall from corrosion current by ACM sensor were defined to consider the rainfall effect. The rain start was showed over 1 μA, the rain end when the rate of change for corrosion current from the start rainfall became 1/2 times or less than 1 μA after 10 minutes. ~~”
Question 6. Figure 2 is cluttered with results but what is the setup of the experimental rig? Shown in figure 1.
- In order to easily explain and to help readers understand better, Figure 2 added the process of prediction method. Each step was explained in manuscript as follow.
“First, the corrosion damage was evaluated using the weight loss method depending on exposure period. Second, the output of ACM sensor was converted to consider the environmental effect such as water vapor and accumulated corrosion current was calculated by the ACM sensor. Third, the relationship equation was used to compare the mean corrosion depth and corrosion current in the same corrosion environment and exposure period. Finally, the mean corrosion depth was estimated by calculating corrosion current during the expected period.”
Question 7. At line 121 [3. Acceleration corrosion test and results], at line 129 it is mentioned that salt spray test were conducted but which one for example ASTM B117? Also salt test are used for aggressive corrosion tests, where is atmospheric corrosion is no an aggressive form of corrosion, unless the air around us has salt a very high salt content. Also the authors didn’t mention how the salt spray and the study of atmospheric corrosion included in this paper are related?
- As your comment, the design of weathering steel bridges excludes the application of a protective coating, and thus the steel is permitted to rust at an uncontrolled rate. However, when in contact with airborne salt, anti-freezing agents, moisture and fugitive dust, the surfaces of the members do not form a protective rust layer in steel structural members. The corrosivity of the structural member and the time-dependent corrosion behavior are important factors in ensuring that weathering steel bridges can be used safely while remaining economical. For example, such environments in Japan are interpreted to correspond to the value of airborne salt deposition rate less than 0.05 mg-NaCl/dm2/day (mdd) as measured by the dry gauze method. The manuscript was explained with acceleration corrosion test condition. The weathering steel does corrode at a lesser rate than carbon steel, however, the more important distinction is that for the marine environment, the rate never plateaus and corrosion continues through for the industrial and rural environments. In these two environments the rate of corrosion stabilizes to a very low corrosion rate of approximately 0.0254 mm/year(0.3 mils/year). An acceleration corrosion test condition was explained in manuscript as follow.
“In the conducted acceleration corrosion test [21], uncoated carbon steel and weathering steel plates were applied to evaluate the corrosion damage according to JWTCS1001 specification. The JWTCS1001 test method evaluates the corrosion resistance of low-carbon steel in an atmospheric environment with relatively low airborne salt.”
In its current form the paper present some basic literature about corrosion, research methodology and results but unfortunately does not show novelty of the research and could not explain how three different modes of corrosion are related. Authors are encouraged to improve the quality and resubmit.
- The corrosion currents were measured by ACM sensor during acceleration corrosion test, its corrosion currents were compared with mean corrosion depth of carbon steel and weathering steel. The equation for relation with mean corrosion depth and corrosion current was proposed in each steel grade. The predicted mean corrosion depths were calculated by proposed equation, and then predicted mean corrosion depths were compared in tested mean corrosion depths. It was validity that method for estimation of time-dependent corrosion depth of carbon and weathering steel using ACM sensor. Finally, the mean corrosion depth that was predicted by ACM for 1 to 10 years from calculated long-term corrosion depths of each steel specimen under acceleration corrosion test environments.

Reviewer 3 Report
The paper discribes the performed tests very well but to use only 100 days experimental data for 10 years corrosion prediction is not correct especially the test was performed as accelerated cyclic test. There is not shown results of ACM sensors´corrosion rate in real atmospheric environment to be compared with accelerated results. On Fig. 9 it is evident that the obtained data covers only first months of steel exposure.
Author Response
The paper describes the performed tests very well but to use only 100 days experimental data for 10 years corrosion prediction is not correct especially the test was performed as accelerated cyclic test. There is not shown results of ACM sensors´ corrosion rate in real atmospheric environment to be compared with accelerated results. On Fig. 9 it is evident that the obtained data covers only first months of steel exposure.
- The field examinations give reliable damage data at the built site or the test site and statistical analysis of the data makes it possible to evaluate the corrosion damage. However, the examinations take from several years to tens of years. In previous study, the corrosion rates of weathering steels exposed in atmospheres with exposure time from 7years to 22years under rural, urban and industrial environments. The corrosion rate of weathering steel is approximately 0.006 mm/year under non-marine atmospheres with corrosivity. In this study, a rapid way to specify the mean corrosion depth of carbon and weathering steel, acceleration corrosion tests were conducted at short-term period. The mean corrosion depth of carbon and weathering steel was evaluated approximately 0.15 mm by acceleration corrosion test. Its results were compared that exposure period 15 years under marine environment over 0.5 mdd. To focus the method for estimation of time-dependent corrosion depth of carbon and weathering, the corrosion currents were evaluated at short-term period using ACM sensor.
In addition, the corrosion current of real atmospheric environment is measuring the long-term period over 2 years. Further, the corrosion currents and mean corrosion depths of acceleration corrosion test will compare the those of real atmospheric corrosion environment.
Author | Corrosion environment | Steel grade | Air-borne sea salt (mdd) | Corrosion rate (mm/year) |
aKamimura | Marine | Weathering steel | 0.01 | 0.001-0.003 |
0.1 | 0.001-0.003 | |||
>0.5 | 0.001-0.003 | |||
bKawabata | Marine | Carbon steel | 0.23 | 0.006 |
0.8 | 0.006 | |||
Weathering steel | 0.23 | 0.006 | ||
0.8 | 0.006 |
a. T. Kamimura, S. Hara, H. Miyuki, M. Yamahita, H. Uchida, Corrosion Science 48 (2006) 2709–2812.
b. F. Kawabata, K. Matsui, T. Obinata, T. Komori, M. Takemura, T. Kubo, Steel plates for bridge use and their application technologies, JFE Technical Report No. 2, JFE Steel, Japan, 2004.

Round 2
Reviewer 2 Report
Unfortunately to my understanding this paper still needs a major re-write before publication. In its current form it cannot be published.
The abstract is still generic and fails to explain what is the significant and which results are important. It’s merely about ACM-Type corrosion sensors’ data.
As mentioned in my previous comments the discussion about the research work was vague, it needs a clearer explanation. For example what was the set-up of the test rig or how it was conducted? The figures are not clear, for example figure 2. There is not enough literature what the authors would like to show to the readers. Again in Figure 2a, what the authors would like to show? There are phase A & B, but how these experimental set-up were conducted? Also page 4 line 128-129, “The results of the acceleration corrosion test that previously were used “what does this sentence mean?
Most of the conclusion is a repetition of the abstract, it needs to be revised completely.
Author Response
Reviewer #2
The abstract is still generic and fails to explain what is the significant and which results are important. It’s merely about ACM-Type corrosion sensors’ data. As mentioned in my previous comments the discussion about the research work was vague, it needs a clearer explanation.
- To improve the manuscript in response, English was corrected by a native speaker.
Question 1. For example what was the set-up of the test rig or how it was conducted? The figures are not clear, for example figure 2. There is not enough literature what the authors would like to show to the readers. Again in Figure 2a, what the authors would like to show? There are phase A & B, but how these experimental set-up were conducted?
-In order to help readers comprehension, the experimental method was more clearly modified in manuscript.
“The test method was combined in the Phase-A cycle with salt spray and the Phase-B cycle without salt spray according to the specification [21]. The salt accumulated at the surface of the specimen in the Phase-A cycle, and, in the Phase-B cycle, corrosion was accelerated to repeat the dry and wet conditions. The Phase-A cycle consisted of 0.5-h wet periods under air stream at 35 °C and relative humidity 90%, 0.5-h fog cycle using a solution of 1 wt.%, 1-h wet periods under air stream at 35 °C and relative humidity 90%, and 6.0-h dry periods under air stream at 40 °C and relative humidity 50%. The Phase-B cycle consisted of 2.0-h wet periods under air stream at 35 °C and relative humidity 90%, and 6.0-h dry periods under air stream at 35 °C and relative humidity 90%. Acceleration corrosion tests were conducted repeatedly in the Phase-A cycle for 144 h and subsequently in the Phase-B cycle for 96 h in a Q-FOG CRH chamber. The corrosion environment applied during the accelerated corrosion tests is summarized in Figure 3 (a). The conducted accelerated corrosion test was performed for 10, 30, 50, and 100 days [20].”
Question 2. Also page 4 line 128-129, “The results of the acceleration corrosion test that previously were used “what does this sentence mean?
In previous study, characteristic of corrosion product and corrosion damage for the carbon and weathering steel were evaluated such as XRD, EDX. “The results of the acceleration corrosion test that previously were used” means the refer to results of corrosion damage in previous study. The manuscript was modified as following
“To review and compare the proposed corrosion damage prediction method, as shown in Figure 3, conducted acceleration corrosion test results were used [20].”

Reviewer 3 Report
No any other comments.
Author Response
Thank you for your helpful comments.